# A Systematic Review of Short-Term Outcomes of Minimally Invasive Thoracoscopic Surgery for Lung Cancer after Neoadjuvant Systemic Therapy

**DOI:** 10.3390/cancers15153908

**Published:** 2023-08-01

**Authors:** Shaina Sedighim, Madelyn I. Frank, Olivia Heutlinger, Carlin Lee, Stephanie J. Hachey, Hari B. Keshava

**Affiliations:** 1Department of General Surgery, Irvine School of Medicine, University of California, 3800 Chapman Ave, Suite, 6200, Orange, CA 92868, USA; 2Irvine School of Medicine, University of California, Orange, CA 92868, USAoheutlin@hs.uci.edu (O.H.); 3Department of Molecular and Cell Biology, Irvine School of Biological Sciences, University of California, Orange, CA 92868, USA; 4Division of Thoracic Surgery, Irvine School of Medicine, University of California, Orange, CA 92868, USA

**Keywords:** non-small cell lung cancer, video-assisted thoracoscopic surgery, neoadjuvant treatment

## Abstract

**Simple Summary:**

Non-small cell lung cancers (NSCLCs) can be treated with chemotherapy, radiotherapy, immunotherapy, or a combination before undergoing surgical resection. However, uncertainty remains regarding the overall outcomes of patients undergoing minimally invasive surgical (MIS) resection of the lung, following systemic treatment. As such, we studied the existing data on outcomes of this patient population via a comprehensive dive into the literature and performed a meta-analysis. Our findings indicate that MIS can be safely performed following systemic treatment in patients with NSCLC. Notably, MIS resection offers added benefits including significantly higher lymph node yield compared to open surgery. We also address the various complication rates of both open and MIS surgical approaches, underscoring the importance of surgeon preparedness to convert MIS approaches to open surgeries in precarious circumstances. Overall, this study provides valuable insights into the safety and efficacy of MIS resection post-systemic treatment for patients with NSCLC, offering potential improvements in patient outcomes and guidance to surgical decision making.

**Abstract:**

Background: Minimally invasive surgeries for non-small cell lung cancers (NSCLCs) such as video-assisted thoracoscopic surgeries (VATSs) and robotic-assisted thoracoscopic surgeries (RATSs) have become standard of care for patients needing surgical resection in early stages. The role for neoadjuvant systemic therapy has increased with patients receiving neoadjuvant systemic chemotherapy and immunotherapy. However, there has been some equipoise over the intraoperative and overall outcomes for these patients. Here, we review the current data regarding outcomes of patients undergoing minimally invasive thoracic surgical resection after systemic chemotherapy, immunotherapy, or both. Methods: A systematic literature review of randomized controlled trials and observational studies presenting data on patients with NSCLC that underwent neoadjuvant systemic therapy followed by minimally invasive surgery was performed assessing complications, conversion rates, and lymph node yield. Results: Our search strategy and review of references resulted in 239 publications to screen with 88 full texts assessed and 21 studies included in our final review. VATS had a statistically significant higher lymph node yield in five studies. The reported conversion rates ranged from 0 to 54%. Dense adhesions, bleeding, and difficult anatomy were the most common reported reasons for conversion to open surgeries. The most common complications between both groups were prolonged air leak, arrythmia, and pneumonia. VATS was found to have significantly fewer complications in three papers. Conclusions: The current literature supports VATS as safe and feasible for patients with NSCLC after neoadjuvant systemic treatment. Surgeons should remain prepared to convert to open surgeries in those patients with dense adhesions and bleeding risk.

## 1. Introduction

For the past few decades, lung cancer has been the leading cause of cancer-related death in the United States with non-small cell lung cancer (NSCLC) accounting for around 85% of all lung cancer cases [1]. While the majority of lung cancer is detected at a late stage, the prevalence of NSCLC has been increasing in line with early-stage lung cancers due to the earlier detection of lung cancer [2,3]. Additionally, advances have been made in effective available treatments. Developments in surgery, chemotherapy, immunotherapy, and radiotherapy lead to a decrease in the incidence of late-stage NSCLC and overall fatality. 

Historically, surgery has been the primary treatment modality for early NSCLC, and definitive chemoradiation has served as the primary treatment for unresectable disease processes. For patients that present with more advanced disease at the time of diagnosis, neoadjuvant chemo- and immunotherapy has been theorized to treat micro-metastatic disease early and improve future ability to resect via downstaging [4]. Response to neoadjuvant treatment may also act to identify patients that will later benefit from adjuvant therapy. Multiple studies have shown the potential for neoadjuvant treatment to improve outcomes for patients with advanced neoplasms. A meta-analysis of 15 randomized controlled trials showed that preoperative chemotherapy was correlated with a 13% relative risk reduction in overall survival. In addition, neoadjuvant chemotherapy prolonged time to distant recurrence and recurrence-free survival in patients with resectable NSCLC [5]. Early data emerging from clinical trials on neoadjuvant immunotherapy suggest that biologics alone or in combination with chemotherapy may significantly improve overall survival, increase cure rates, and reduce recurrence rates [6]. 

Minimally invasive (MIS) techniques such as video-assisted thoracoscopic surgery (VATS) and robotic-assisted thoracoscopic surgery (RATS) have become the standard of care over open thoracotomies for early-stage lung cancer [7,8]. The advantages of minimally invasive surgery include shorter hospital stays and fewer postoperative complications [9]. However, in cases of more advanced neoplasms that require chemo- or immunotherapy [10] prior to surgery, the implementation of minimally invasive approaches has been controversial. Some surgeons report that neoadjuvant systemic therapy creates a less ideal operative field due to hilar fibrosis and bleeding risk, thus recommending open approaches as the safer alternative [11]. As such, surgeons have been hesitant to utilize VATS for oncological surgeries in this patient population due to the fear of sacrificing quality lymph node dissection and exposing patients to increased perioperative risk. Many investigations have emerged in order to identify the balance between the benefits of VATS and neoadjuvant treatment with the perceived risks of performing VATS following chemotherapy, radiotherapy, or immunotherapy. Recently, more studies have demonstrated the feasibility of minimally invasive approaches for the treatment of patients with NSCLC who have undergone neoadjuvant systemic therapy [12,13,14].

In this article, we review the current data regarding outcomes of patients undergoing minimally invasive thoracic surgical resection after systemic chemotherapy, immunotherapy, or both. We focus on operative complications, conversion rates, and the overall safety of those patients undergoing VATS or RATS resection compared to their counterparts who underwent open approaches.

## 2. Methods

A systematic literature review was performed in line with the PRISMA checklist. The following search strategy was utilized and sourced from PubMed and relevant references: (non-small cell lung cancer) AND (neoadjuvant chemotherapy OR neoadjuvant immunotherapy) AND (VATS OR video-assisted thoracoscopic surgery OR minimally invasive surgery OR robotic surgery). A search was conducted on the Cochrane Central Register of Controlled Trials with the following strategy: non-small cell lung cancer in All Text AND neoadjuvant in Title Abstract Keyword AND surgery in Title Abstract Keyword—with Cochrane Library publication date in the last 2 years, in Trials (Word variations have been searched). Inclusion criteria consisted of English-language randomized controlled trials and observational studies presenting data on patients with NSCLC that underwent neoadjuvant therapy followed by minimally invasive surgery. 

We excluded (1) case studies and reviews, (2) studies on staging or mediastinoscopy, (3) those that did not separate results by neoadjuvant treatment or surgical approach, and (4) ongoing trial data. Papers were screened by one reviewer, and two authors reviewed the full texts for inclusion.

Publications were managed and screened using Covidence software. Extracted variables included stage, type of neoadjuvant treatment, amount of radiation, type of surgery, and time to surgery. Outcomes extracted include: rate of open surgery or VATS (including RATS), rate of conversion to open surgery, operative complications, and lymph node resection. Cause of conversion from VATS to open surgery was recorded. The number of lymph nodes harvested in VATS and open cases was collected along with significant differences if reported.

## 3. Results

Our search strategy and review of references resulted in 239 publications: among these, 6 duplicates were removed, 88 full texts were assessed, and 21 studies were included in our final review (Figure 1). The studies consisted of Phase 1, 2, and 3 trials, prospective observational, retrospective observational, and database studies from 2016 to 2022 (Table 1). Two papers included patients with stage IV NSCLC, while others included stage I–IIIB (T1-4, N0-2, M0) [13,15]. Types of neoadjuvant therapy included chemotherapy (dual-platinum based chemotherapy), immunotherapy (including PD-1 inhibitors and monoclonal antibodies), radiotherapy (40–60 Gy), and a combination. Immunotherapy treatments included nivolumab, sintilimab, erlontinib, paxopanib, atezolizumab, pembrolizumab, carmrelizumab, durvalumab, toripalimab, and tisleeizumab. Two papers included chemotherapy, two included immunotherapy, six included chemo- and radiotherapy, nine included chemo- and immunotherapy, and three included three therapies.

Time to surgery was reported by nine publications. The mean time to surgery ranged from 26.5 to 93 days. Yang reported a significant difference in time to surgery between those receiving open surgery with a mean of 90 days and a range of 78–111 versus VATS with a mean of 93 days and a range of 77–114 (*p* < 0.01) [22]. Zhang reported that of the patients that had over 42 days between neoadjuvant treatment and surgery, most (76%) were due to adverse events from neoadjuvant therapy, and 24% were due to economic reasons [33].

Four papers included only the VATS approach, and the majority included both VATS and open approaches. Three papers also included RATS as a subset of VATS [17,25,29]. Of the papers that included both open and VATS, most patients received an open surgery (Table 1).

Of the 12 publications that reported the number of lymph nodes resected, five found statistically significantly more lymph nodes resected with an open approach compared to VATS (Table 2). Other publications found no significant difference between the two approaches.

The reported conversion rate from VATS to open ranged between 0 and 54% with the exception of one paper that excluded converted cases [18] and one paper that did not report conversion rate [21].

Nine papers reported the reason for conversion from VATS to open. Dense adhesions were the cause in 39 cases, bleeding and metastatic or fibrous lymph nodes were the cause in 12 cases each, and difficult anatomy/dissection was the cause in 4 cases. One paper reported 19 cases of conversion due to primary tumor invasion [33].

There were zero 30-day mortalities reported in 10 publications, and 4 did not report on mortality. No significant differences by surgical approach were found across those that noted one or greater patient death [12,14,15,19,22,26,29,30].

Operative time was compared by surgical approach in 6 publications [7,13,18,20,26,30]. Zhang found significantly shorter operative times within the MIS group (160 ± 40.4 vs. 177.7 ± 57.7, *p* = 0.042) [7]. Matsuoka noted statistically significant shorter operations in MIS group as well; however, the times were not reported [20].

Estimated blood loss was found to be decreased in the MIS group by Kamel (100cc (50–150) vs. 150cc (100–250), *p* = 0.02), Tian (100cc (30–200) vs. 200cc (300–525), *p* < 0.001), and Zhang (149cc (±57.9) vs. 321.2cc (±72.3), *p* = 0.021) [13,30,35]. Matsuoka reported significantly less estimated blood loss but did not provide numerical data [20]. Fang did not find any statistically significant difference in estimated blood loss [18]. Yang found no difference by surgical approach in bleeding requiring transfusion or reoperation [16].

Of the 10 papers that reported statistical comparisons of complications between approaches, three found significant differences with a *p* < 0.05. Compared to the open approach, VATS had fewer respiratory and cardiac complications [25], medical complications [26], or major complications [28]. The most common complications across both groups were prolonged air leak, arrythmia including atrial fibrillation, and pneumonia (Table 3, Appendix A). 

All papers concluded that VATS is safe and feasible for patients with NSCLC after neoadjuvant chemotherapy, radiotherapy, immunotherapy, or a combination of treatments.

## 4. Discussion

Since the adoption of minimally invasive surgery (MIS) into the repertoire of thoracic surgery, MIS has become increasingly prevalent for both benign and malignant diseases. The approach has been shown to have a multitude of advantages including reduced surgical trauma, decreased postoperative pain, shorter length of hospitalizations, and faster recovery times [36]. MIS has revolutionized the field of thoracic surgery and is now the major modality for surgery in many practices that operate on early-stage lung cancer without neoadjuvant therapy. However, its role in patients who have undergone neoadjuvant cancer treatments for NSCLC remains an area of controversy. 

Chemotherapy and immunotherapy have been increasingly implemented in the neoadjuvant treatment regimen in patients with locally advanced NSCLC. In comparison to adjuvant therapy, neoadjuvant treatment has the potential to increase the number of resectable tumors, improve surgical margins, decrease the size of resection, and promptly treat micro-metastatic disease. Additionally, compliance with systemic therapy is thought to be superior in the neoadjuvant setting, as post-surgical patients may not tolerate treatment should they have difficulty recovering from surgery. Compared to surgery alone, neoadjuvant chemotherapy with subsequent surgery has been shown to improve survival in patients with early-stage NSCLC especially in patients with stage II and III [35]. Overall, the utility of neoadjuvant treatment encompasses transforming unresectable tumors into resectable tumors and reducing the risk of metastatic disease in those patients with lymph node metastasis and higher stage [37]. 

Operative safety, the ability to adequately visualize mediastinal structures, and lymph node yield are at the forefront of surgeons’ minds when contemplating minimally invasive versus open approaches in patients who have undergone neoadjuvant therapy. Safety and visualization are critical to ensuring an adequate surgery is performed with minimal complications. Thus, some surgeons choose to avoid MIS in later stages of malignancy, as mediastinal structures are reportedly distorted after receiving neoadjuvant chemotherapy and/or immunotherapy. These surgeons state that such distortion could lead to increased difficulty and complexity of lung and lymph node resection, which can prevent an oncologically sound operation. However, over the past few years, surgeon experience with video-assisted thoracoscopic surgery (VATS) is increasing along with the adoption of RATS and in addition to improvement in intraoperative technology and visualization. Therefore, minimally invasive approaches have been proven to be just as safe as open thoracotomy in patients who have undergone neoadjuvant systemic therapy.

In order to ensure patient safety and operative ease, there are instances in which surgeons must convert from a MIS to an open approach. In our review, we found a wide range of conversion rates from MIS to open (0–53%). This finding is consistent with a retrospective National Cancer Database analysis that found neoadjuvant chemotherapy to be associated with a higher conversion rate of 18.3% than those without neoadjuvant treatment with a *p* < 0.01. In contrast, Muslim et al. found neoadjuvant treatment was not an independent predictor of conversion [38]. Comparatively, a meta-analysis of early-stage NSCLC without neoadjuvant treatment found a median conversion rate of 8.1% (range 0–15.7%) [8]. The most common reasons for conversion were dense adhesions, bleeding, and fibrotic lymph nodes. Studies that reported multiple cases converted for adhesions attributed the fibrosis as a response to immunotherapy treatment [15,17,33]. Indeed, histological features of response to immune checkpoint blockade have been shown to present as increased lymphocytic infiltrate and fibrosis, which will be discussed later in this article [39]. Positive nodes and late stage were also associated with conversion. Additionally, studies acknowledged more complicated pulmonary resections to be associated with conversion rate [33]. While converted operations have similar mortality rates to non-converted cases, they can be associated with greater morbidity. Complication rates, chest tube duration, and length of hospital stays are all greater in patients who were converted from MIS to open compared to those who were not [40]. However, for optimal safety and to prevent worse morbidity or an incomplete resection, conversion may be unavoidable. 

Another imperative point of comparison involves perioperative complications. Our review found that the most common complications in MIS and open approaches included prolonged air leak, arrythmia including atrial fibrillation, and pneumonia. Interestingly, all papers found MIS to have comparable or fewer complications compared to patients undergoing open surgery. Investigators have shown that those undergoing MIS after neoadjuvant therapy have a lower rate of postoperative pain and lower volume of chest tube drainage compared to the thoracotomy group [18,26]. Unsurprisingly, multiple studies report the shortened length of hospital stay in patients undergoing MIS versus thoracotomy [26,41].

Intraoperative systematic hilar and mediastinal lymph node assessment in early-stage NSCLC is necessary for proper staging accuracy, detection of early metastasis, and better survival [42,43]. Intraoperative systematic lymph node assessment entails either mediastinal lymph node sampling or mediastinal lymph node dissection. Randomized controlled trials have illustrated conflicting data with regard to survival differences between these two techniques. The American College of Surgeons Oncology Group Z0030 found no survival difference between systematic surgical mediastinal lymph node sampling and mediastinal lymph node dissection [44]. Contrarily, Wu et al. found that patients who underwent mediastinal lymph node dissection had a better survival and were more likely to find occult N2 disease compared to mediastinal lymph node sampling [45]. Other studies have implicated the number of lymph node dissected to be an independent prognostic factor of staging accuracy and possibly survival benefit [46,47]. Specifically, patients with accurately assessed mediastinal lymph nodes are more likely to find occult N2 disease steering patients to much-needed adjuvant systemic therapy. Taking into account all of these data, in 2020, the American College of Surgeons Commission on Cancer released intraoperative standard 5.8 [48] for pulmonary resection stating that at least one named or numbered hilar lymph node and at least three named or numbered mediastinal lymph nodes should be resected and identified in the pathology report. Accurate lymph node assessment is imperative to understand patients’ recurrence risk and to help guide post-operative treatment in patients with NSCLC. As such, selecting the appropriate surgical approach to allow for accurate lymph node assessment is a key component in patient prognosis and survival. 

Through our review of patients receiving neoadjuvant treatment for NSCLC in the literature, there appears to be a discernible difference of lymph node yield and approach to surgery after neoadjuvant therapy. We found four studies that reported a statistically significant higher lymph node resection in the open surgery over MIS. That being said, these studies noted comparable oncological efficacy despite there being a lower number of resected nodes with MIS [26,33], the same number of stations resected, equivalent upstaging [18], and comparable survival [13]. Additionally, more recent studies including Kamel et al. noted an greater number in number of nodes resected via MIS with increased surgeon experience [13]. One publication reported significantly more nodes resected via MIS [19]; however, this was attributed to the open group receiving more pre-operative video-assisted mediastinoscopy dissections. Tian et al. found no difference in evaluating the extent of lymph node resection and recurrence in 127 patients who underwent MIS or thoracotomy resection of locally advanced NSCLC after neoadjuvant therapy [30]. The authors found that there was no difference in the number of lymph nodes dissected, lymph node stations sampled, and recurrence-free survival in these groups.

Among minimally invasive techniques, there are reported differences in lymph node yield. Toker et al. compared the effectiveness of lymph node dissection in the open, VATS and RATS approaches [49]. In a retrospective analysis, they found that RATS yielded significantly more lymph nodes resected in total and significantly more N1-level nodes. RATS was particularly more effective in harvesting station #11 and #12 lymph nodes compared to open thoracotomy and even VATS. Lastly, Nachira et al. showed no difference in nodal upstaging between uniportal MIS and open thoracotomy [50].

There is increasing support that the lymph nodes themselves are affected by the neoadjuvant systemic treatment. Pathologists not only report residual viable tumors in the lymph node and primary site but also assess treatment effect and fibrosis. For this reason, it can be difficult to truly determine the number of lymph nodes, illustrating the need for very accurate hilar and mediastinal lymph node station assessment noted by the COC Intraoperative standard 5.8 [48].

Preoperative radiation has been debated, and many clinicians opt for solely chemotherapy and immunotherapy as the mainstays for neoadjuvant treatment. However there are multiple phase II trials that report higher response rates, more pathologic response, and increased downstaging with the inclusion of radiotherapy to the neoadjuvant regimen [51,52]. That said, more postoperative complications have been reported with preoperative chemoradiotherapy compared to chemotherapy alone [51]. 

Radiation treatment in particular is associated with an increased conversion rate due to the subsequent edema, dense adhesions, and fibrotic changes. Huang et al. reported a 37.5% conversion rate where all but one patient received preoperative radiotherapy [12]. The only perioperative death in that study was a patient with sequential chemoradiotherapy complicated by radiation esophagitis. Yang et al. reported that one in seven converted cases had received pre-operative radiotherapy [16]. In line with the understanding that radiation less than 40 Gy may prevent operative complexity, pre-operative RT with a Gy of 50–60 has been associated with conversion [15]. The increased conversion rates and higher ratio of initial open approaches for patients that received radiation in our review suggest that neoadjuvant radiation at high doses may not be compatible with MIS. 

Immunotherapy, specifically immune checkpoint inhibitors, has shown great promise in improving overall survival in both metastatic and locally advanced NSCLC patients [37]. 

A multitude of studies have shown increased overall survival and disease-free survival in the metastatic setting when utilizing immunotherapy along with chemotherapy rather than chemotherapy alone. The PACIFIC trial illustrated that immunotherapy increases survival in early-stage unresectable lung cancer with the addition of Durvulumab after chemoradiotherapy [53]. For resectable patients, the use of systemic therapy with the inclusion of immunotherapy is being studied with promising data both in the neoadjuvant and adjuvant setting. IMpower 010 showed that adjuvant atezolizumab after adjuvant chemotherapy increased disease-free survival especially for patients with 1% or higher PD-L1 expression. Neoadjuvant chemotherapy and immunotherapy have been shown beneficial for disease-free survival with the recent findings in CHECKMATE 816 and NADIM I and II [34,54]. However, questions still remain regarding the timing of the immunotherapy regimen relative to surgical resection. 

In an attempt to describe the utility of nivolumab immunotherapy as a supplement to neoadjuvant chemotherapy, Forde et al. conducted a recent phase III clinical trial (Checkmate 816) comparing those receiving standard of care neoadjuvant chemotherapy versus those receiving neoadjuvant chemoimmunotherapy [34]. Relevant to our review, the authors note that the addition of neoadjuvant immunotherapy did not increase the incidence of adverse events or feasibility of surgery with many patients undergoing a minimally invasive surgery. Similarly, Provencio et al. conducted a phase II, multi-center study (NADIM) to observe outcomes for stage IIIa NSCLC patients treated with neoadjuvant chemoimmunotherapy followed by surgical resection [55]. The treatment was well tolerated and led to the complete resection in all patients who underwent surgery with no delays in surgery related to their neoadjuvant therapy.

Utilizing immune checkpoint blockade in the neoadjuvant setting likely provides an early opportunity to enhance the patient’s immune response against their tumor, as the tumor is left in situ, allowing for the larger response. However, this response may make the future resection more difficult due to the newly unleashed anti-tumor immunity that serves as a principal driver of organ inflammation and fibrosis [17].

Although well tolerated in the majority of cases, immune checkpoint blockade can result in side effects known as immune-related adverse events (irAEs), which can affect multiple organ systems, including the lungs [56]. PD-L1 serves a vital role in suppressing inflammatory responses through PD-1 mediated T-cell inhibition, and treatment with anti-PD-1 monoclonal antibodies (such as nivolumab) may lead to inflammatory toxicity by reactivating T cells that are normally held in check by interactions with PD-L1 [57,58]. In rare cases, immune-mediated lung injury in response to immune checkpoint blockade can manifest as checkpoint inhibitor pneumonitis (CIP), which is a severe pulmonary irAE that can lead to the excessive formation of fibrous connective tissue in the lungs [59]. Due to the scarring and thickening of the lung tissue, CIP-associated fibrosis can lead to decreased lung function that may impede surgical interventions, increase the risks associated with surgery, and/or lead to surgical complications. CIP is also associated with an increased risk of pneumonia. In a study by Wu et al., a total of 6360 subjects treated with anti-PD-1 (nivolumab or pembrolizumab) from 16 phase II/III clinical trials were pooled for meta-analysis to evaluate the incidence and risk of PD-1 inhibitors-related pneumonitis in patients with NSCLC, melanoma, RCC, and other cancers [60]. For all malignancies, the overall incidence of pneumonitis during anti-PD-1 immunotherapy was 2.92% (95% CI: 2.18–3.90%) for all-grade and 1.53% (95% CI:1.15–2.04%) for high-grade pneumonitis. Notably, patients with NSCLC are at higher risk of pneumonitis when compared across different tumor types, with an overall incidence of all- and high-grade pneumonitis of 4.27% (95% CI: 3.26–5.58%) and 2.04% (95% CI: 1.37–3.03%), respectively [60]. Treatment with PD-1 inhibitors was associated with a significantly increased risk of pneumonitis compared with routine chemotherapy. However, the risk of PD-1-induced pneumonitis was found to be dose-independent with no significant differences in risk between high- and low-dose anti-PD-1 [60].

While rare, pneumonitis should be carefully considered when evaluating patients for surgery if administering immune checkpoint blockade in the neoadjuvant setting. Yet detecting checkpoint inhibitor pneumonitis, and accurately predicting post-operative outcomes, prior to surgical intervention has remained challenging. A CT radiomics-based predictive model was recently shown to provide reasonable predictions of immunotherapy-associated pneumonitis in tumors treated with PD-L1 inhibitors (AUC. 0.74, 95% CI: 0.53–0.95) [61]. While promising, further development and clinical validation is needed to achieve a reliable predictive model for immune checkpoint blockade response, which is specifically focused on pulmonary toxicities such as pneumonitis that may lead to poor surgical outcomes. Immunotoxicity has previously been linked to rapid diversification of the T-cell infiltrate immediately following immune checkpoint blockade, although the pathogenesis of pneumonitis remains poorly understood. In a recent study reporting on the perioperative outcomes of 21 patients who underwent anatomic lung resection post-treatment with immune checkpoint blockade, diffusing capacity for carbon monoxide (DLCO) was the only factor associated with postoperative complications [62]. Surgical complications were significantly associated with patients who had lower pre-operative DLCO values compared to patients who did not experience post-operative complications (61.0% (57.0–67.0%) vs. 90.5% (72.0–105%); *p* = 0.027); odds ratio 0.095 (95% confidential interval (95% CI): 0.014–0.809) [62]. Lower DLCO values can be caused by fibrosis, suggesting a mechanistic link between irAEs and post-operative complications. Additional research should be conducted on a larger cohort to determine if pre-operative DLCO values are predictive of poorer surgical outcomes generally. Ultimately, biomarkers are needed to guide clinical and surgical management decisions in this patient population to ensure the patient can tolerate surgery and improve outcomes.

With the increased implementation of immunotherapy in lung cancers, surgeons should be aware of the increased risk of fibrosis and not have any hesitation to start with minimally invasive surgery. That being said, surgeons should be ready to convert to open surgery to not compromise an oncologically sound operation after immunotherapy. Over the next few years, further studies being conducted along with the increased use of neoadjuvant immunotherapy will truly give us an idea of the adoption, complications, and conversion rates of minimally invasive techniques with the use of neoadjuvant immunotherapy outside of clinical trials. 

While the strengths of this review include the number of randomized-controlled trials included, broad search strategy, and systematic screening of papers, the limitations include the inclusion of observational studies that impart risk of bias and the lack of true randomized controlled trials in this setting. Future reviews will be needed as ongoing trials of neoadjuvant treatment for NSCLC utilizing MIS approach are released. Additional studies could report further details on reasons for conversion and complications along with the increased use of robotics in patients following neoadjuvant therapy.

## 5. Conclusions

Minimally invasive thoracic surgery (MIS) has become the standard of care for lung cancer resection due to faster postoperative recovery. It has been studied extensively in cases amenable to upfront resection. However, limited data exist with regard to the efficacy of minimally invasive thoracic surgery following neoadjuvant therapy. In this discussion, we summarize the existing data on VATS and robotic surgery following neoadjuvant therapy. The extent of lymphadenectomy and feasibility of resection following neoadjuvant therapy will always be at the forefront of these discussions, as they are critical to determining the overall prognosis of the patient. More studies will be required to determine what is the best approach to surgical resection following neoadjuvant therapy.

## Figures and Tables

**Figure 1 cancers-15-03908-f001:**
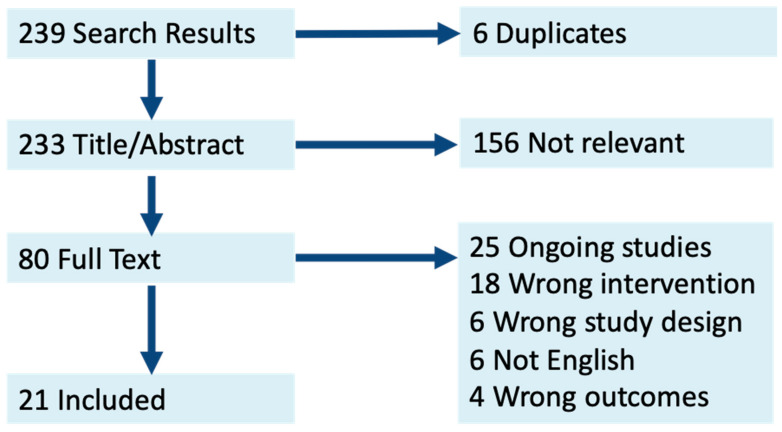
Literature search PRISMA diagram.

**Table 1 cancers-15-03908-t001:** Table of all study outcomes.

Author	Year	Stage	Neoadjuvant	Open N(%)	VATS N(%)	Conversion N(%)
Huang [12]	2013	IIA-IIIB	CT, IT, RT	0 (0%)	42 (100%)	7 (17%)
Yang, C [16]	2016	IA-IV	CT, RT	203 (74%)	69 (25%)	7 (10%)
Kamel [13]	2017	I-IV	CT, IT	74 (64%)	40 (35%)	5 (13%)
Bott [17]	2018	I-IIIA	IT	7 (35%)	10 (50%)	7 (54%)
Fang [18]	2018	IIB-IIIB	CT	67 (80%)	14 (17%)	NR
Jeon [19]	2018	IIIA	CT, RT	18 (54%)	17 (48%)	5 (28%)
Matsuoka [20]	2018	NR	CT, RT	31 (28%)	79 (72%)	4 (5%)
Yang, Z [15]	2018	IIB-IIIB	CT	0 (0%)	29 (100%)	1 (3%)
Shu [21]	2020	IB-IIIA	CT, IT	14 (53%)	12 (46%)	NR
Yang, C [22]	2020	NR	CT, RT	2221 (76%)	676 (23%)	152 (22%)
Duan [23]	2021	IIA-IIIB	CT, IT	4 (17%)	14 (61%)	2 (9%)
RomeroRoman [24]	2021	IIIA	CT, IT	20 (48%)	21 (51%)	4 (19%)
Cabanero sanchez [25]	2022	NR	CT, IT, RT	135 (51%)	74 (28%)	21 (8%)
Dell’Amore [26]	2022	IIA-IIIB	CT, IT	93 (60%)	62 (40%)	8 (5%)
Deng [27]	2022	IIIB	CT, IT	0 (0%)	31 (100%)	0 (0)
Jeon [28]	2022	IIIA	CT, RT	350 (90%)	35 (9%)	6 (17%)
Kamel [29]	2022	NR	CT, IT, RT	7894 (70%)	2753 (24%)	557 (16%)
Tian [30]	2022	IIB-IIIA	CT, RT	71 (56%)	56 (44%)	6 (11%)
Tong [31]	2022	IB-IIIA	IT	2 (8%)	18 (72%)	5 (20)
Yao [32]	2022	IIIA-IIIB	CT, IT	0 (0%)	11 (100%)	1 (9%)
Zhang [33]	2022	IB-IIIB	CT, IT	78 (59%)	53 (40%)	42 (54%)
Forde [34]	2022	IB-IIIA	CT, IT	173 (70%)	73 (26%)	38 (13%)

VATS = video-assisted thoracoscopic surgery, CT = chemotherapy, IT = immunotherapy, RT = radiotherapy, NR = not reported.

**Table 2 cancers-15-03908-t002:** Lymph nodes.

Author	Year	LN Open (Median N)	LN VATS (Median N)	*p*-Value
Huang [12]	2013	NR	16.88	
Kamel [13]	2017	15	12	0.945
Fang [18]	2018	20	16	0.011
Jeon [19]	2018	13.5	24	0.004
Yang, Z [15]	2018	NR	21.9	
Yang, C [22]	2020	11	12	0.38
Dell’Amore [26]	2022	26	20	0.022
Deng [27]	2022	NR	16	
Jeon [28]	2022	22.5	22	0.217
Kamel [29]	2022	10	9	<0.001
Tian [30]	2022	19	17	0.337
Zhang [33]	2022	23	19.5	0.013

LN = lymph node, VATS = video-assisted thoracoscopic surgery, NR = not reported.

**Table 3 cancers-15-03908-t003:** Aggregate complication frequency.

	Open	VATS
Complication	Freq Range (%)	Freq Range (%)
Prolonged air leak	0–22	0–12
Arrythmia	0–22	0–23
Pneumonia	0–10	0–6
Wound infection	0–11	0
Cardiac complication	0–14	0–3
Atelectasis	0–6	0–6
ARDS	0–6	0–6
Pneumothorax or effusion	0–1	0
Fistula	0–1	0
Empyema	0–1	0–3
Pulmonary Embolism	0–1	0–12
Respiratory failure	0–3	0
Chylothorax	0–1	0

VATS = video-assisted thoracoscopic surgery, Freq = frequency, ARDS = acute respiratory distress syndrome.

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
