# Peer review of "A Systematic Review of Short-Term Outcomes of Minimally Invasive Thoracoscopic Surgery for Lung Cancer after Neoadjuvant Systemic Therapy"

_cancers, 2023, doi:10.3390/cancers15153908_

Round 1

Reviewer 1 Report

I am pleased to have the opportunity to review this paper.

The authors reported a systematic review of short-term outcomes of a minimally invasive surgical approach for non-small cell lung cancer after neoadjuvant systemic therapy.

I think the systematic review was performed in a reasonable manner.

However, the reported results are quite poor informative to discuss the safety and efficacy of the minimally invasive approach.

I would request to add information regarding the following important perioperative outcomes: mortality and morbidity rate in each approach, operation time, and blood loss during operation.

Also, there are several minor points requiring revision.

Line238  “much needed” is duplicated.

Line313  stageIIIa should be stageIIIA

Author Response

Dear reviewer, 

Thank you for taking the time to read over our manuscript and provide critical feedback on our work. Your time is much appreciated. 

Please see a point-by-point reply to the aforementioned comments: 

"I would request to add information regarding the following important perioperative outcomes: mortality and morbidity rate in each approach, operation time, and blood loss during operation". - This review addresses the short-term outcomes of the MIS patient population. However, we do appreciate that mortality is an important factor to report. As such, we have amended the results section to report these results. We have additionally included EBL and operative time. Please refer to lines 156-168 of the new manuscript. The remainder of points addressing morbidity is included throughout the results section. 

Also, there are several minor points requiring revision.

Line238  “much needed” is duplicated.

Line313  stageIIIa should be stageIIIA

  • We have made these corrections, thank you.

Thank you again for your time and consideration. We are enthusiastic that our changes have addressed all concerning points and hope they suffice for acceptance for publication. 

Reviewer 2 Report

Dear Editor and Authors,

Thank you kindly for asking me to review this manuscript titled “A Systematic Review of Short Term Outcomes of Minimally Invasive Thoracoscopic Surgery for Lung Cancer after Neoadjuvant Systemic Therapy” by Dr. Sedighim and colleagues from the University of California, Irvine School of Medicine.

In this systematic review the authors present the currently available literature on minimally invasive surgery (VATS or RATS) for NSCLC in patients having had neoadjuvant chemotherapy. This is quite an interesting question to tackle considering the hesitance (which the authors very fittingly mentioned) in the thoracic surgical community to perform minimally invasive surgery in this patient cohort due to fear of increased risk and complications and due to inadequate lymphandenectomy!

In total this is a very well conducted review with a clear and robust methodology, well written and structured and easily understood. The commentary on the discussion section is applicable and relevant.

Two very minor points I would like to raise is:

1.       Table 4 is quite compressed and “busy” and I am not sure how useful it is. Therefore, I would suggest it be removed as it does not significantly add to the overall paper.

2.       Throughout the text the authors utilize the term VATS however as the authors have mentioned a number of studies included both VATS and RATS therefore maybe a better term to utilize throughout the text would be Minimally Invasive Surgery (MIS).

In conclusion, I think this is a valuable and interesting review which I feel should be presented in the thoracic surgical community.

Good quality language. Nothing to add.

Author Response

Dear reviewer, 

Thank you for taking the time to read over our manuscript and provide critical feedback on our work. Your time is much appreciated. 

Please see a point-by-point reply to the aforementioned comments: 

1.       Table 4 is quite compressed and “busy” and I am not sure how useful it is. Therefore, I would suggest it be removed as it does not significantly add to the overall paper. - We agree with this assessment and have made Table 4 a Supplemental figure. Thank you. 

2.       Throughout the text the authors utilize the term VATS however as the authors have mentioned a number of studies included both VATS and RATS therefore maybe a better term to utilize throughout the text would be Minimally Invasive Surgery (MIS). - We agree with this and have made the appropriate adjustments. 

Thank you again for your time and consideration. We are enthusiastic that our changes have addressed all concerning points and hope they suffice for acceptance for publication.